# Silver-Nanowire-Based Elastic Conductors: Preparation Processes and Substrate Adhesion

**DOI:** 10.3390/polym15061545

**Published:** 2023-03-21

**Authors:** Kai Yu, Tian He

**Affiliations:** College of Mechanical and Electrical Engineering, Qingdao University, Qingdao 266071, China

**Keywords:** AgNW, elastomer, adhesion, substrate, soft robot

## Abstract

The production of flexible electronic systems includes stretchable electrical interconnections and flexible electronic components, promoting the research and development of flexible conductors and stretchable conductive materials with large bending deformation or torsion resistance. Silver nanowires have the advantages of high conductivity, good transparency and flexibility in the development of flexible electronic products. In order to further prepare system-level flexible systems (such as autonomous full-software robots, etc.), it is necessary to focus on the conductivity of the system’s composite conductor and the robustness of the system at the physical level. In terms of conductor preparation processes and substrate adhesion strategies, the more commonly used solutions are selected. Four kinds of elastic preparation processes (pretensioned/geometrically topological matrix, conductive fiber, aerogel composite, mixed percolation dopant) and five kinds of processes (coating, embedding, changing surface energy, chemical bond and force, adjusting tension and diffusion) to enhance the adhesion of composite conductors using silver nanowires as current-carrying channel substrates were reviewed. It is recommended to use the preparation process of mixed percolation doping and the adhesion mode of embedding/chemical bonding under non-special conditions. Developments in 3D printing and soft robots are also discussed.

## 1. Introduction

Flexible screens, wearable sensing devices, soft robots and other fields are developing rapidly, but rigid electronic components and high-cost and high-complexity processes such as stereolithography may not be compatible with them, which drives more and more researchers to enter into low-cost flexible electronic products composed of stretchable conductors, such as RF circuits, flexible antennas, transistors, sensors, etc. [1,2,3,4]. In the process of manufacturing flexible electronic components, the most important thing is the stretchability of the electrical interconnection, which also promotes the research and development of flexible wires or stretchable conductive materials. Flexible stretchable conductors are the basis of flexible electrical interconnection, and their basic requirement is to manufacture a high conductive path with large deformation, bending and torsional resistance.

Optional materials for flexible conductors include malleable metal networks [5], liquid metal [6,7,8,9], carbon nanotubes [10], metal matrix composites [11,12,13,14], hydrogels polymers [15,16], etc. The types of conductors indicate that most of these materials evolved from metals like silver, gold and copper, or are composed of new materials like carbon-based materials or gels. The stretchable network is made of metals with low tensile properties, and is not suitable for the internal circuit of flexible electronic devices or even soft robots. Liquid metal has good mechanical properties and is not prone to fracture failure due to its fluidity. However, its melting point is relatively high. For example, the melting point of gallium indium alloy (EGaIn) is 15 °C. Carbon nanotubes have the advantages of lightness and good tensile performance, but their conductive performance is often difficult to break through 1 × 10${}^{4}$ S/m. In practical use, the circuit loss is too large, which is not conducive to large-scale promotion. Graphene also has the same problem. Hydrogel polymers have excellent tensile properties and have a certain self-healing ability. However, when used alone, although hydrogels can be regarded as ionic conductors, they still face the problem of poor electrical conductivity.

The electrical conductors of metal matrix composite materials include metal nanoparticles, metal nanowires, metal slurry, etc. The metals are mainly gold, silver and copper, as the electrical conductivity of other metals is much lower than these three metals. As the metal with the best electrical conductivity, silver is undoubtedly the best potential candidate. In actual manufacturing, it not only has far better electrical conductivity than graphene [17] and higher heat dissipation than carbon nanotubes [18], but it is also stronger than the existing commercial indium tin oxide (ITO) electrode [19]. Among silver matrix composites, silver nanowires (AgNWs) have become a more prominent material in recent years. Compared with silver paste, AgNW has a lower preparation temperature [20] and can be used as a long conductor to connect silver nanoparticles or silver sheets, further improving the continuity and stability of conductive paths [21]. It can improve the conductivity instead of nanoparticles and reduce the resistance change caused by deformation by its certain flexibility. It can also be prepared into a layered network of multilayer silver nanowires to overcome the problems of grooves, holes and high roughness of single-layer silver nanowires [22], indicating that silver nanowires are the most promising materials for the development of flexible circuits.

There have been a lot of studies on the preparation of silver nanowires. From the perspective of chemical synthesis methods, these include the template method [23,24,25,26], electrochemical method [27], wet chemical method [28,29], alcohol reduction method [30,31,32], solvothermal method [33,34], photoreduction method [35], etc. From the perspective of additives, a large number of influential factors such as halogen ion content [36] and coating agent concentration (PVP or a monoalcohol copolymer-based system [37]) have been studied. Reaction temperature, reaction time and other basic constants are unnecessary. However, for practical applications, the main research directions are conductive films (electrodes) [1,38,39,40], sensors [41,42], fabric [43,44], radio frequency [45], filter elements [46], shielding devices [47] or further memory [48]. What is outstanding is the condition or single use of receiving external signals (pressure, temperature, etc.), while there are still few studies on sending signals such as power supply coils, wireless energy transmission, signal interconnection, etc. In terms of the base material of the research support, polyethylene terephthalate (PET) and polydimethylsiloxane (PDMS) [49], which occupy the majority, are completely sufficient for the tensile ratio of human wearable devices (about 40%), but the potential of high tensile performance is still small (elongation at break: PET about 120%, PDMS about 100%), and for high ductility and bending fatigue, interface deformation and adhesion still need to be further studied. Therefore, starting from the vision of preparing system-level flexible systems, based on silver nanowires as the main material, we investigated about 1600 relevant studies from the past 8 years and a small number of early representative studies in terms of elastomers, conductivity, tensile properties, adhesion, printing, etc., and obtained more than 100 documents with strong correlation with AgNW by taking the highest tensile rate greater than or equal to 100% while maintaining conductivity as the screening criterion. In this paper, the preparation of elastomers and the adhesion enhancements with elastic substrates are reviewed. Section 2 of this paper introduces four kinds of elastic conductive preparation processes including silver nanowires. Section 3 describes the adhesion mechanism of five kinds of conductive networks and substrates. Section 4 describes the current achievements of silver nanowire conductive networks in the fields of 3D printing and robotics, and Section 5 summarizes the prospects and suggestions for building fully autonomous flexible agents.

## 2. Fabrication of Elastic Bodies Containing Silver Nanowires

Silver nanowires themselves have a certain degree of flexibility and can even produce directivity under the scraping of Mayer rods. However, they do not have good tensile properties after synthesis and separation. A randomly distributed nanowire network can only withstand small strains, and is prone to cracking or even fracturing when subjected to tensile, torsional and other stresses, resulting in a surge in resistance. Although we can alleviate part of the problem by increasing the strength of silver nanowires [50], if there is no flexible material added at all, it is difficult to bear small-radius bending and large tensile deformation in flexible electronic devices or soft robots. Therefore, it is necessary to make composite elastic materials by adding highly elastic materials while using silver nanowires as conductors. Four elastomer preparation methods are reviewed, and those whose maximum tensile rate is lower than 100% while maintaining electrical conductivity are not discussed.

### 2.1. Prestretching/Geometric Topological Matrix

By prestretching flexible materials or through geometric topology design, a substrate with certain tensile properties can be obtained, and then a stretchable conductor can be obtained by adding silver nanowires. The predrawing process consists of a three-step process in which the elastomers are prestretched and then deposited into a conductive path, after which the strain is released and its pretension variable is used to offset the strain to be applied externally. Aliphatic aromatic random copolyester (Ecoflex) is a material with excellent elasticity (maximum strain > 900%) [51]. The two-dimensional prism-type Ecoflex [52] deposits AgNW after stretching (Figure 1), and the two-dimensional elastic elongation can reach 750% after releasing the strain. Cui Zheng’s team [53] explained the relationship between surface folds and tensivity in the manufacturing process of two-dimensional deposition using the printing method, and fabricated a flexible LED circuit using the PDMS substrate. Geometric shape design can prepare microstructures resistant to tensile and bending deformation, usually by folding or bending shapes. The horseshoe design [54] or the snakelike cross structure of styrene-butadiene-styrene block copolymer (SBS) (Figure 2) [55] can resist large three-dimensional deformation. For deformation beyond the pre-designed condition, conductive materials can be added to make up for the rupture gap of the conductive seepage network caused by excessive stretching, such as silver nanoparticles [56], single-walled carbon nanotubes (SWCNTs) [57], etc. This method is very common, but it has the problem of low extensibility (<200%), the direction of strain resistance is generally specific, and the unconsidered direction is often poorly tensile, so it needs to expand its application in the condition of free deformation.

### 2.2. Conductive Fibers

For one-dimensional stretching or small-radius bending deformation, conductive fibers can be prepared via the electrospinning method [58]. Generally, polyurethane (PU) and SBS are used to provide tensile properties. PU can be stretched to form a wavy surface before deposition of AgNW by combining the directional prestretching method [59], so as to increase the deposited surface area and improve the stretchability and small-radius (0.3 mm) flexural property. Wet spinning provides a mature idea for the preparation of silver nanowires as conductive elastomers preserved in liquid phase. For highly elastic materials with hydrophobicity, additives can be used to enhance the miscibility of the conductive phase and the elastic phase. For example, by adding polyethylene glycol monomethyl ether thioctylate (TA-mPEG) [60] and combining hydrophilic silver nanowires with dithioyl groups to achieve a solubility similar to PU and other substrates, the binding uniformity and tightness of the conductor and elastomer are improved to form a composite conductive fiber with a strain of 200%. N, N-dimethylformamide (DMF) was used for ligand exchange with polyethylpyrrolidone (PVP), a commonly used coating agent [61], to achieve a high miscibility with SBS and obtain a highly elastic fiber with a maximum conductive tensile ratio of 220%(fracture tensile ratio of 900%). For the large tensile deformation of elastic fibers, it is very easy for the conductive network to break when the value is greater than 150%. By using the principle of prestrain, a double-coated cotton yarn (DCY) [62] can be helically wound on the elastic matrix, and after dipping in silver nanowire solution and H${}_{2}$ plasma treatment, PDMS can be used for sealing and encapsulation (Figure 3). When stretched, cotton yarn with a spiral structure assumes most of the stress and plays the role of a flexible spring. It can achieve 6.88 × 10${}^{4}$ S/m conductivity and excellent bending resistance under 500% strain. Moreover, by virtue of the low toxicity of PDMS, it can realize the manufacture of subcutaneously biocompatible circuits in mice. The composite fiber can be used to weave fabrics and prepare large-area three-dimensional strain signal corresponding flexible sensors [63], which proves the application prospect of solid and stretchable composite fibers as electrical wiring systems in large-area stretchable electronic equipment.

### 2.3. Aerogel Composites

The aerogel preparation method can be used in the case of extremely light conductor requirements, and its high surface area characteristics of multi-holes can improve the permeability of elastomers and nanowires, and have better adaptability to 3D deformation. Both volume and shape can be molded. Based on an anisotropic silver nanowire solution, Jung et al. [64] converted the solution from a semi-diluted state to an isotropic concentrated state through low-temperature compression to achieve wet gel preparation. Then, the network was assembled via the supercritical drying method and dried in a solvent without surface tension. A basic aerogel with a density of 8.8 × 10${}^{4}$ g/m${}^{3}$ and conductivity of 3 × 10${}^{6}$ S/m was obtained, which provided an experimental basis for the preparation of a silver nanowire aerogel. Sterling silver nanowire aerogel is brittle and has a high-porosity mechanical response. The stress will rise abruptly during large deformation, so it is necessary to strengthen the elasticity. This problem can be solved via the freeze-drying method [65,66], supplemented by PDMS filling the conductive network. Through subsequent heat treatment, composite conductors with tensile properties of more than 130% can be obtained. Qian et al. [67] studied the differences between two methods combining freeze casting, freeze drying and heat treatment, and found that the holes obtained during the liquid nitrogen freeze-drying method with isotropic inward solidification and convergence are uniform. The changes in seepage conductivity and stress in the network were summarized in detail, and an aerogel with a conductivity of 5.1 × 10${}^{4}$ S/m was obtained. However, it is undeniable that the tensile property of an aerogel without mechanical reinforcement is low, and the elasticity enhancement is not obvious.

**Figure 3 polymers-15-01545-f003:**
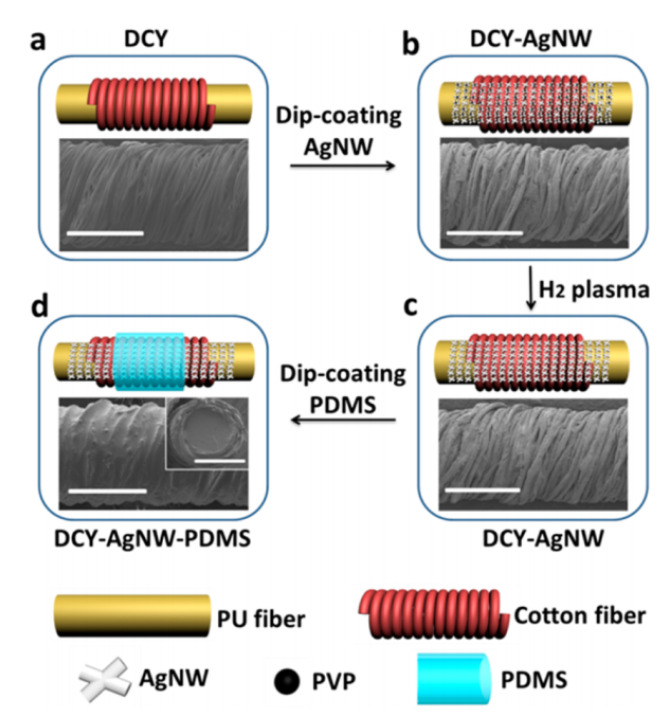
Schematic illustration of the fabrication process of the composite fiber. (**a**) Structural representation and SEM image of the DCY. For simplicity and clarity, only one CY was demonstrated in the structural drawing. (**b**) Structural representation and SEM image of DCY-AgNW before H${}_{2}$ plasma treatment. (**c**) Structural representation and SEM image of DCY-AgNW after H${}_{2}$ plasma treatment. (**d**) Structural representation and SEM images of DCY-AgNW-PDMS. The inset SEM image shows the cross-sectional structure. The PDMS infiltrated the whole CY layer. Scale bar for all images: 500 μm. DCY, DCY-AgNW and DCY-AgNW-PDMS all have diameters of 650 μm. Reprinted with permission from [62]. Copyright 2015 American Chemical Society.

For large deformation and self-healing requirements, the use of a more flexible hydrogel reinforcement strategy is a better choice. Dual-network hydrogels can dissipate strain energy effectively by deforming the network conformation or sliding cross-junctions along the polymer chain. Mechanical reinforcement using poly2-acrylamide-2-methylpropanesulfonic acid-polyacrylamide (PAMPS-PAAm) hydrogels [68] can reduce tearing stresses by nearly 20 times. Composite nanoreinforced hydrogels provide ideas that can increase elasticity. Cong’s [69] team used N,N’ -bis (acrylyl) cystamine (BACA) and a silver nanowire aerogel to form Ag-S bonds (Figure 4). After modification, poly (n-isopropylacrylamide) was filled and strain was shared through the deformation of each unit. The network was maintained stably through the fracture recombination of the Ag-S bond, thus maintaining excellent electrical conductivity under an 800% strain cycle and self-recovery under infrared irradiation, which proves the potential of this method in the manufacture of flexible stretchable electronic devices.

### 2.4. Mixed Seepage Dopant

The conductive phase of silver nanowires can be directly mixed with the elastic phase in proportion, and a highly conductive elastomer can be obtained after curing under the condition of uniform dispersion. This method requires that the seepage threshold of the conductive network should be as high as possible, so two solutions are extended. One uses similar surface energy materials to enhance the degree of crosslinking between the elastomer and the nanowire network.

In the silver nanowires prepared by mainstream alcohol reduction method, the surface energies of AgNWs and PVP are about 0.081 J/m${}^{2}$ and 0.078 J/m${}^{2}$, respectively, which are significantly different from those of rubber and SBS with surface energies of 0.020 J/m${}^{2}$–0.040 J/m${}^{2}$ [70]. The surface energy of thermoplastic PU (TPU) is 0.056 J/m${}^{2}$, which is much closer. Zhu et al. prepared inks at 60 °C with a ratio of AgNW:TPU ≈ 1:9, and obtained a high conductivity of 3.668 × 10${}^{5}$ S/m and high adhesion of more than 20 machine washes in a single printing without a seal on the fabric. Moreover, the fabric could withstand more than 200% of stretching, making it very stable. The high responsiveness, good stability and low lag of PU and poly(3, 4-vinyl dioxyethiophene): poly(styrene sulfonate) (PEDOT:PSS) and AgNW three-component composites have also been investigated [71]. Weak interactions between components can cause PEDOT:PSS polymer chains to slip or break during stretching. Thus, AgNW becomes the so-called “rigid component” indirectly, reducing cracking and obtaining more than 240% extensibility.

For PDMS and SBS with low surface energy, they can be crosslinked by additives to achieve more uniform mixing with silver nanowires. Vinyl-sealed PDMS can undergo hydrosilylation reactions with polymethylhydrosiloxane (PHMS) [72] to obtain three-dimensional cross-linked networks with excellent tensile properties. After doped silver nanowires are cured at 393 K, a self-defined shape conductive body can be obtained, and a stable path can be maintained under a tensile rate of more than 800%. For the conductive barrier problem caused by PVP residue, Choi et al. [73] used hexylamine and PVP to achieve ligand replacement, mixed silver nanowires in a gold sheath with SBS and toluene, obtained conductors enriched in conductive and elastic phases after drying and forming (Figure 5), and characterized the significant effect after the increase of the conductive percolation threshold. After hot-pressing treatment, the tensile ratio was as high as 840%, and the conductivity was as high as 7.6 × 10${}^{6}$ S/m under the appropriate ratio. In addition, the high biocompatibility brought by the gold sheath expands the application, creating new possibilities in the field of soft bioelectronics.

A comparison of different preparation methods is shown in Table 1:

## 3. Adhesion Principle of Silver Nanowires to Substrate

After the preparation of silver nanowire elastomers, apart from the factors related to polymer materials and silver nanowire density, such as electrical conductivity, a big problem is often encountered in practical use; that is, the low adhesion between silver nanowires and flexible substrates [74,75]. The surface properties of metal and polymer are so different that even if silver nanowires are sprayed on the substrate, it is difficult to withstand even a simple wipe to fall off the surface and completely fail. Even if not completely shed, holes and defects in the conductive network will often appear after stretching, resulting in loss of conductivity [76]. The most basic method involves using silver nanowires with the highest aspect ratio to have more advantages under complex strain [77]. The scheme of the adhesion principle between silver nanowires and the substrate is worth studying. In order to solve this problem, many researchers have found corresponding solutions for their respective substrates and materials. According to the principle, it is divided into five types. Only flexible base materials are discussed, and hard bases or paper bases such as glass are not discussed.

### 3.1. Coating Method

The silver nanowire network can be covered with a layer of material with good adhesion to the substrate, so that the silver nanowire network is fixed on the substrate and acts as a protective layer to prevent the silver nanowire from being oxidized.

Zinc oxide is low-cost, non-toxic and has a wide range of sources. It is prepared in solution [78] and coated on the top of silver nanowires on a PET substrate. After annealing, a relatively dense low-junction resistance network can be obtained, which can withstand 1000 instances of repeated bending with a radius of 5 mm to maintain good adhesion, and can also withstand the Sellotape test. Magnetic-assisted electrodeposition is another way of covering via the secondary deposition of metallic layers on silver nanowires. Zhang et al. [79] took thermal stability as the starting point, selected nickel as the covering layer and PET as the base, and formed a strip-covering layer tens of nanometers thick with the tuning effect of a rotating magnetic field, which could withstand 200 instances of tape-stripping and a 400 °C high temperature (>30 min), and also improved the heat resistance. The coating of metals and their oxides needs to consider the compatibility of their rigid and flexible substrates under motion, as well as their suitability to the equipment in the use environment. The use of acrylate composite coatings has little impact on transparency and tensile properties [80], and can achieve hydrogen bonding with substrates such as PMMA. Spraying can achieve good adhesion, and is more suitable for complex deformation.

### 3.2. Embedding Method

#### 3.2.1. Reverse-Layer Processing

The principle of reverse-layer treatment is to spray or spin-coat the substrate with silver nanowires or its solution, then pour liquid stretchable polymer (generally PDMS), after fully permeating into the silver nanowire network, and then cure or anneal with violet/infrared light to obtain a composite conductor embedded on the polymer surface. This method usually has good adhesion due to the fusion of the conductive network with the polymer.

Xu and Zhu [81] placed suspended AgNW droplets on silicon, glass or plastic substrates, then poured liquid PDMS and cured them (Figure 6). After stripping, a highly crosslinked PDMS-AgNW elastic conductor was obtained, which could guarantee no cracking within 100% tensile strain, and its stable conductivity could reach 5285 S/cm, which is a typical preparation process. Curing the base after pre-tensile casting [82] can effectively improve the electrical stability and mechanical robustness. In order to enhance the penetration between the polymer and the conductive network, PDMS pressurized penetration can be performed on the substrate by means of the vacuum pressure difference [83] or spraying [84] to improve the embedding effect. This method can also be used to manufacture flexible batteries that can stretch and twist [85], providing conditions for the manufacture of embedded active circuits. Curing methods are usually UV/IR irradiation or annealing. Because the melting point of silver nanowires is lower than that of silver, the annealing process is equivalent to the welding treatment of joint points to a certain extent, so as to improve the conductivity. Zhang et al. [86] added PVDF solution to AgNW film, and the transparent film obtained by reannealing could resist the grinding of 2000 mesh sandpaper. It is worth pointing out that Zhou et al. [87] embedded the silver nanowire network and MXene on the PC substrate into the PVA film via the method of spin-coating and hot pressing, and the combination was equally strong.

#### 3.2.2. Sintering Method

The resistance of the conductive channel of composite materials is generally divided into three types [41]: contact resistance, bulk resistance and tunneling resistance. For the naturally settled conductive network, improving the conductivity can reduce the resistance as much as possible by enhancing the contact between nanowires. The low ablative flux threshold of AgNWs allows direct laser etching to penetrate the network, generally without damaging the substrate [88]. High-intensity pulsed light (HIPL) uses the heat of instantaneous sintering to instantly melt the silver nanowires with the shallow layer of the substrate, which sinks deeper than the heavy silver and thus integrates with the substrate. At the same time, high heat can instantly raise the temperature of the conductive network to hundreds or even thousands of degrees, which is equivalent to rewelding the relatively relaxed network, while removing the possible residual PVP on the surface, and obtaining a denser high-conductivity path, which is a fast and mass processing method. Jiu et al. [89] applied high-intensity pulsed optical sintering (Figure 7) with a length of 50 μs to AgNWs on a PET substrate to obtain a film that can withstand a peel resistance of 19 square resistances of 3M tape. Song et al. [90] sintered silver nanowires on PI, PET and PEN substrates with an intense pulsed light duration of 500 μs, and obtained embedded conductive networks with strong adhesion, which confirmed the universality of this process for flexible polymer substrates. Yang et al. [91] used HIPL with a length of 50 μs to study the tensile properties of silver nanowires sintered on the surface of a PU substrate, showing that the embedded conductive network has electrical conductivity stability after 1000 instances of 100% stretching. The sintering method can also be completed via a strong pulsed ion beam [92]. The strong ion beam with a time of 200 ns is embedded in the silver nanonetwork on the PET substrate. After 200 bending cycles, the resistance is almost unchanged, and the transparency is as high as 94%, which shows the strong potential of this method in TCF manufacturing.

The sintering method requires attention to the laser irradiation position, application time and rate of high heat pulses. Since the temperature has long exceeded the glass transition temperature (Tg) of most polymers, it is easy to cause severe damage to the base and scrap it. Dai et al. [93] explored in depth the precise control of SPNW of silver nanowires, where the nanowelding is laser-induced through the plasmonic-resonance-enhanced photothermal effect. It was shown that the illumination position is a critical factor for the nanowelding process. Jiu et al. [88] summarized the thermal properties of various polymer substrates and discussed this problem in detail. It can be confirmed that the thermal properties and experimental conditions of each different substrate must be independently considered when using this method, and the damage of the substrate can be controlled in a very small area or very shallow surface (<3.5 μm [92]) as far as possible by adjusting the intensity of the pulse and irradiation time.

### 3.3. Changing the Surface Energy

This method is based on the different properties of metals and substrates. Silver nanowires are hydrophilic, while polymer substrates are often hydrophobic, so it is impossible to produce a good adsorption force solely by surface adsorption of the two parts. Therefore, the adhesion ability of the two parts can be improved by surface modification; that is, changing the surface energy. Surface energy changes can be achieved via ultraviolet/ozone (UVO) treatment, as well as plasma treatment.

UVO treatment can hydroxylate the surface atoms of polymers such as PDMS [94] (Figure 8), resulting in a high density of hydrophilic functional groups on the surface, which can strongly adsorb deposited molecules or atoms, and thus generate a good adsorption capacity with silver nanowires. Pandey et al. [95], combined with pre-stretching, silanized PDMS treated with UVO, and tested the adhesive force with silver nanowires to withstand tape. Lin et al. [96] studied the surface chemical properties and long-term stability of a UVO-treated substrate by measuring XPS, AFM and the contact angle, and found that PMMA, COC and PC treated with UVO experienced hydrophobicity recovery within 4 weeks. The incidence rate depended on the duration of UVO treatment, which posed a challenge to the application of long-term stability. Surface silanization is resistant to high temperature and humidity [17], and is suitable for many bending conditions with good flexibility.

Plasma treatment can also reduce the contact angle of the hydrophobic layer to aqueous solution [97] and enhance hydrophilicity. PEDOT:PSS can be swirled on PET after it is enhanced by oxygen plasma to reduce the surface roughness of silver nanowires [98] and improve the density of the conductive network. Silanization is also applicable to plasma-treated surfaces [99], and can effectively enhance adhesion and durability. For the possibility of residual PVP on the surface, plasmas of inert gases such as argon [74] can be used to treat the surface of the conductive layer, and can retain the conductivity and remove the organic materials on the surface.

### 3.4. Chemical Bonds and Forces

#### 3.4.1. Chemical Bonds

Since silver is a transition metal, most covalent bonds forming when it adheres to the substrate are coordination bonds.

Polydopamine (PDA) is a common modified material for PU. After modifying the polymer surface with catechol and amine groups, arc-pair electrons can be generated to pair with silver nanowires, thus forming covalent bonds when covering silver nanowires and obtaining strong adhesion. Zhang et al. [100] modified the surface of PU sponge with polydopamine and obtained strong adhesion after dipping into an isopropyl alcohol solution of silver nanowires, which could withstand 1000 instances of bending, stretching and torsion without breaking of the conductive path. Li et al. [101] prestretched PU and modified the surface with PDA to achieve uniform distribution of AgNWs and little change in bending resistance. Catechol-type polyaniline (c-PANI) treated with a catalyst [102] can produce an electrostatic complex with 17 catechol groups, conjugated with silver nanowires to form a high-conductivity hybrid network, which is closely bonded with PET.

Benzophenone generates isolated electronic groups and some free radicals in its polymer chain under ultraviolet light, which can form stable covalent bonds with hydrogels, and obtain high adhesion while having a low modulus and strong tensile properties. Kim et al. [103] solidified a hydrogel containing sodium alginate and a silver sheet on a PET/Ecoflex film treated with dipropanone (Figure 9). After UV curing, the conductor did not peel off at an elongation of up to 1780%. Good adhesion stability was achieved. Alginate (AA) is the only polysaccharide containing carboxyl groups in each component residue. A large number of free carboxyl groups coordinate with metal ions in the form of negative ions, promote binding with silver nanowires, and inhibit the formation of voids in the deformation process; these can be used as good additives. Jin et al. [104] modified PET with alginate, and added PDA at the same time, so that the conductive network and the base realized conformational contact; they could resist the 90° peeling of 3M tape more than 100 times, and the adhesion was reliable. PET can also be modified with silane [105]. The silane group forms silanol after encountering a hydrogel, and forms a siloxane covalent bond after UV curing to achieve a reliable connection. In addition, as a chemical bond with generally high binding strength, a covalent bond can realize reversible addition reactions and be used to prepare conductive elastomers with certain healing properties [106]. If combined with a hydrogel, the self-healing composite can withstand the conductive network and the substrate’s relative sliding will be larger. After cyclic strain release, the conductive network often produces agglomeration in the springback process, which damages the conductivity. AA can promote the formation of covalent bonds between silver nanowires and the substrate [107], reduce the slip and tear of the interface between metal and substrate, and reduce the surge in resistance value caused by cyclic strain.

For the substrates exposed to hydroxyl, surface modification of silver nanowires can be performed [108] to achieve covalent bond and electrostatic bond formation simultaneously. Amphiphilic substances such as conjugated polyelectrolyte (CPE) [109] and ethanolamine (Figure 10) [110] can be used to realize the functional connection between the conductive network and the interface of organic substrates. The polar groups can unify the surface as far as possible and promote the generation of ionic bonds, enhance the bonding strength, and have good electrical stability and tensile properties.

#### 3.4.2. Hydrogen Bonding

The force of a hydrogen bond is generally less than that of a chemical bond. However, since the binding force between the metal and the substrate without any adhesion treatment is mainly a van der Waals force, hydrogen bonding can still be regarded as an auxiliary strengthening measure. For silver nanowires prepared via the alcohol reduction method or wet textile method, PVP on the surface can achieve hydrogen bonding with porous elastic polymers such as PU foam [111] to obtain a conductor with a tensile ratio greater than 310% and electrical conductivity of 9.19 × 10${}^{4}$ S/m, which can withstand 1000 tensile cycles of 70% strain and maintain electrical stability. It has been proved that hydrogen bonding can achieve a strong adhesion effect. Polyurethane urea (PUU) exhibits stronger hydrogen bonding than PU [112] due to the presence of two hydrogen donors in the urea group, and can form stronger hydrogen bond connections with carboxylic acid groups on the main chain of AgNWs.

For the enhancement of the tensile property of the substrate, Guo et al. [113] designed a variety of non-covalential crosslinking agents to prepare a PDMS-SS-IP-BNB polymer (Figure 11), which had an average self-healing property of 93%. The intrachain hydrogen bond allows the extension of polymer chains, the interchain hydrogen bond can be repeatedly bonded/broken during the process of chain sliding, and the stronger disulfide bond can be affected by the breaking loss force, thus achieving the maximum elongation of 14,000%. Based on this, it is possible to obtain high conductivity under certain sliding conditions if the AgNW conductive network is combined.

### 3.5. Adjusting Tension and Diffusion

The surface tension of silver nanowires and flexible substrates is different, so the adhesion can be improved by adjusting the tension and promoting the penetration and fusion of the two substrates. Lee et al. [114] used isopropyl alcohol to adjust the evaporation rate of PEDOT:PSS solution coated on the conductive network, and induced tightly bonded nanowires through the capillary force of the conductive polymer solution, thus greatly improving the conductivity at the junction to achieve the effect of nanowelding, which could withstand 2 × 10${}^{4}$ bending cycles. Wang dried AgNW/CNT electrodes slowly in wet air [115] and used capillary-force-induced self-welding (Figure 12) to increase the tensile ratio to 580%, which is of significance for the low-cost development of hybrid elastic circuits.

## 4. Applications in 3D Printing and Soft Robotics

3D printing is a rapid prototyping additive manufacturing technology with quantitative properties, almost no material waste and the advantage of producing almost arbitrary structures. Using 3D printing is a good way to reduce costs and increase production rates for sensor arrays, RF modules and even soft robots that produce silver nanowire substrates. This is because silver nanowires have strong process compatibility and are suitable for electrodriven printing [116], inkjet printing [117], screen printing [118], aerosol jet printing [119] and direct writing [120].

The configuration of silver nanowire inks generally involves a high-boiling-point solvent, because solvents with low boiling points (such as ethanol) are prone to the coffee ring effect, where the nanowires cluster around, creating a gap in the conductive path. High-boiling-point alcohol [121] or isopropyl alcohol/glycol [122] can be used to improve the uniformity of conductive ink during evaporation, but it is difficult to meet the requirements of elasticity. Zhu et al. [70] directly combined silver nanowires with TPU, and obtained a printable conductor using the phase-conversion process of TPU and DMF in methanol. The conductivity was as high as 3.668 × 10${}^{5}$ S/m, and the strong adhesion after printing on textiles provided a direction for the rapid preparation of inkjet printing on stretchable electronic fabrics. AgNW can also be mixed with TPU resins to produce 3D capacitor sensors using digital optical processing (DLP) printing technology [123] while having high electrical conductivity.

Although inkjet printing is an efficient and convenient preparation method, its disadvantage is that the width of the conductive path is determined by the nozzle diameter, and the resolution is difficult to improve. In addition, in order to prevent the blockage of the nozzle, AgNW with a shorter length and diameter is often used to prepare the ink, which makes it difficult to fully exploit the advantage of the high length–diameter ratio of AgNW. Screen-printing can solve this problem to some extent [118] by controlling the mesh/template size to improve the resolution (about 50 μm), and there is no requirement for the aspect ratio. However, for the requirement of real-time control of the wire diameter, it is recommended to use the electrohydrodynamic (EHD) printing process [124] to prepare flexible heaters and electrodes by adjusting the size of the Taylor cone to control the wire diameter in real time. The nozzle diameter can also be appropriately increased to meet the requirements of AgNW printing. For the yellow bias of silver nanowire ink during printing, anionic blue dye [125] can be used to inhibit it.

For soft robots, rigid components limit their flexibility. By 3D printing part of the sensor and the conductive path, the original rigid parts and low efficiency manual preparation processes can be replaced to a certain extent, while improving the flexibility of part of the intelligence. It can be seen that 3D printing technology can have a good development prospect in the field of soft robot preparation.

## 5. Summary and Outlook

Silver nanowires are potentially the most useful and easily realized conductive materials for flexible electrical interconnection. Stretchable and bendable conductors and substrates made from silver nanowires can solve the problem that traditional silicon-based circuits cannot deform greatly and adapt to concave and convex surfaces. Therefore, the preparation of elastic bodies containing silver nanowires and the methods of enhancing adhesion have been reviewed in this study.

The manufacturing process of silver nanowires is mature and they can be mass-produced. Good breakthroughs have been made in the fabrication of electrical conductors containing silver nanowires. It is recommended to use predrawn conductive fibers to prepare large-area conductive fabrics and achieve mixed porous doping. A pretensile matrix is generally used for the case of a fixed direction or predictable deformation, and it is difficult to cope with random strain. A conductive aerogel is generally not used except in the case of strict weight requirements, because of its relatively complex preparation process, long preparation cycle and high cost (such as the supercritical drying method). Conductive fibers can be adapted to strain in all directions, and devices such as sensors can be prepared, which have the potential for further research. The advantage of a dopant is that it can obtain parts with different properties by changing the distribution ratio of each group, the combination of each component is tight, and the shape and structure of the product can be prepared arbitrarily, with good universality. In terms of material selection, combining with Ecoflex, hydrogels and other composite substrates can obtain good tensile properties, which can be considered when large deformation (>500%) is required. For flexible electronic devices with small tensile ratios, such as human wearable devices, which are generally less than 50% [126], PDMS, PET, PU, etc. can be used to achieve this goal.

For enhancing adhesion, embedding and chemical bonding are recommended processes and mature methods. The embedding method combined with a certain degree of the hot-pressing process can significantly improve the conductivity, the sintering process can greatly reduce the resistance, and the damage to the substrate is controllable. Chemical bonding can solve the problem of poor adhesion microscopically and is the most secure way of strengthening. The cladding method often conflicts with the flexibility index in operation. The tension and diffusion method mainly depends on adjustment of the evaporation rate, which is relatively small. The UVO treatment of the substrate can change the macroscopic surface energy, which is a kind of icing on the cake. Of course, the substrate itself still has room for development, and can be further improved in the aspects of self-healing and high resilience [127,128]. Further studies are being conducted on the fatigue of the connected interface and the influence of the deformation after cyclic strain release on the performance of the electrical network [129].

As a new generation of processing technology, 3D printing is also a process path worth studying. Compared with the methods described above, the outstanding advantage is that the quantity of ink can be provided according to the need to avoid waste and strong controllability, which is of great significance for the large-scale promotion of the conductive path-processing of silver nanowires. Water-based or alcohol-based silver nanowire composite inks can be combined with spray, drip and other methods for layer thickness controllable superposition and multi-component mixed printing, which has the advantages of easier quantization and batch production compared with spin-coating and other processes.

Fully autonomous soft robots are very interesting products. With the development of rope-driven and pneumatic soft robots, intelligent and interactive soft robot systems will become the innovation point and future development direction [130]. Highly robust audio-photoelectric integrated interactive displays [131], protein-elastomer hybrid artificial photoelectric skin [132] and multifunctional touch screens [133] have also begun to be developed. The core of a fully autonomous soft robot is the flexible interconnection of system-level circuits. A highly conductive composite silver nanowire ink can realize the complete interconnection of distributed components such as chips, sensor units and antennas on a flexible substrate through 3D printing technology. If flexible sensors, electrodes [134] and antennas can be used, the flexibility of the whole circuit can be realized, so that the flexible electronics can develop towards system-level interconnection.

## Figures and Tables

**Figure 1 polymers-15-01545-f001:**
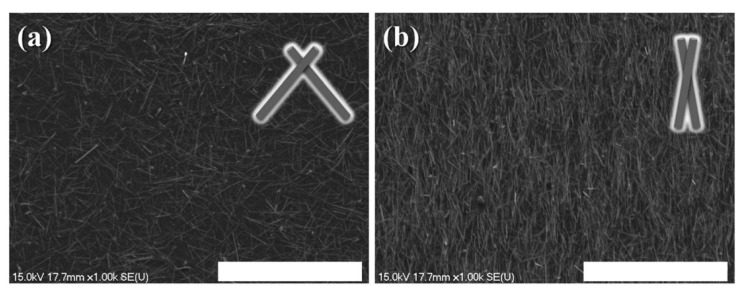
SEM image of final mesh heater before (**a**) and after (**b**) biaxial stretching of serpentine structure; stretching rate 100% (scale bars: 50 μm). Reprinted with permission from [52]. Copyright 2016 RSC Publishing.

**Figure 2 polymers-15-01545-f002:**
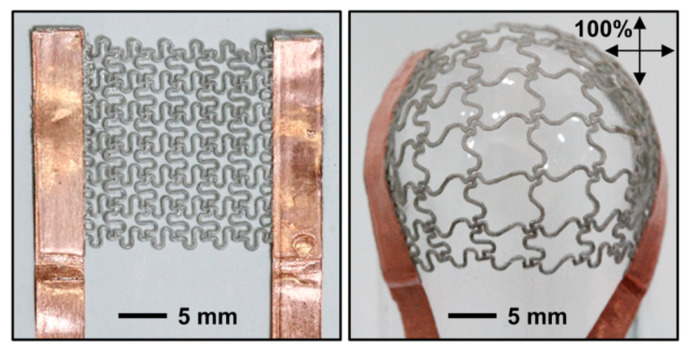
Optical image of a mesh heater (**left**) before and (**right**) after biaxial stretching up to 100%. Reprinted (adapted) with permission from [55]. Copyright 2015 American Chemical Society.

**Figure 4 polymers-15-01545-f004:**
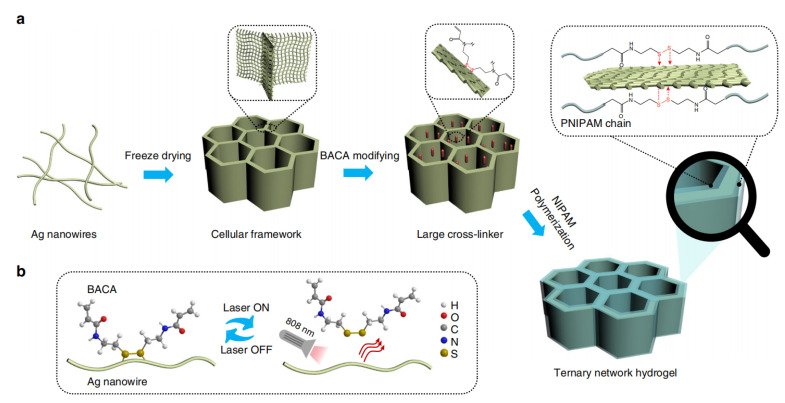
Struture design and self-healing mechanism. (**a**) Schematic illustrations of the preparation of AATN hydrogel. Typically, AgNW aerogel with cellular structure was prefabricated by the freeze-drying method. Being modified with BACA molecules, AgNW aerogel was used as large cross-linker for the synthesis of AATN hydrogel. (**b**) Schematics of the dynamic Ag-RS interaction between BACA and AgNWs under the NIR laser on/off. Reprinted with permission from [69]. Copyright 2018 Springer Nature.

**Figure 5 polymers-15-01545-f005:**
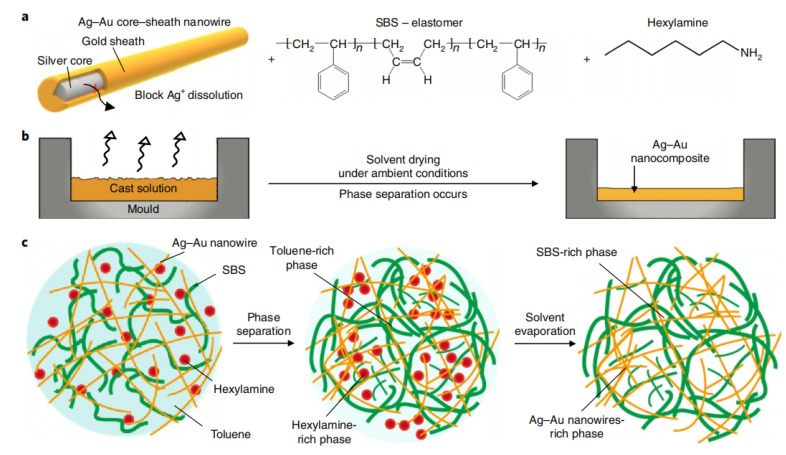
Preparation of Ag-Au nanocomposites. (**a**) Ag-Au nanocomposites are composed of a mixture of Ag-Au nanowires modified with hexamine ligands, SBS elastomers and another hexamine in toluene. (**b**) Diagram of solvent-drying process under environmental conditions. (**c**) The initial solution (**left**) is separated into an Ag-Au-rich nanowire phase and an SBS-rich phase (**middle**) during dry casting. The micro-structured Ag-Au nanowire nanocomposites are formed by solvent evaporation (**right**). Reprinted with permission from [73]. Copyright 2018 Springer Nature.

**Figure 6 polymers-15-01545-f006:**
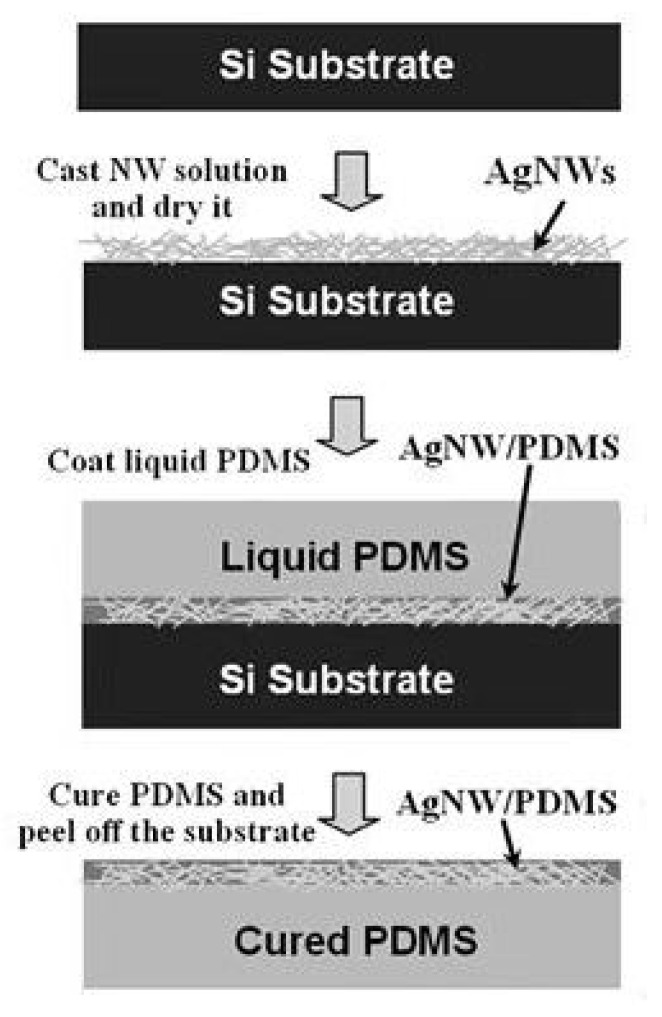
Schematic showing the fabrication process of AgNW/PDMS stretchable conductors. Reprinted with permission from [81]. Copyright 2012 John Wiley and Sons.

**Figure 7 polymers-15-01545-f007:**
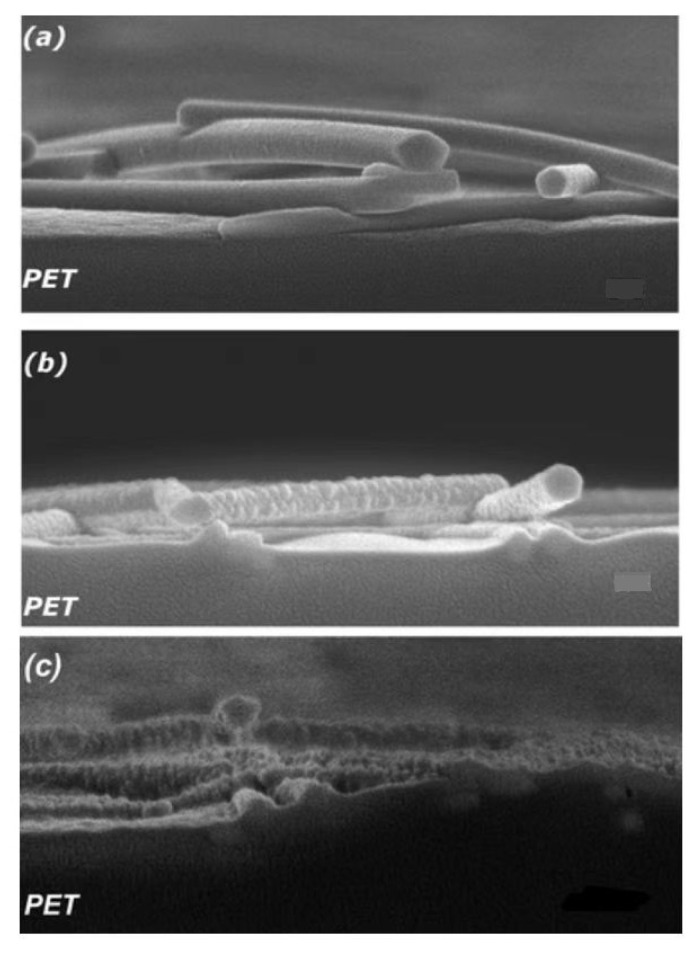
Cross-sectional SEM images of AgNW films on PET substrates (**a**) before and (**b**,**c**) after HIPL sintering with light intensities of 11,400 J/m${}^{2}$ and 23,300 J/m${}^{2}$, respectively. Reprinted with permission from [89]. Copyright 2012 Royal Society of Chemistry.

**Figure 8 polymers-15-01545-f008:**
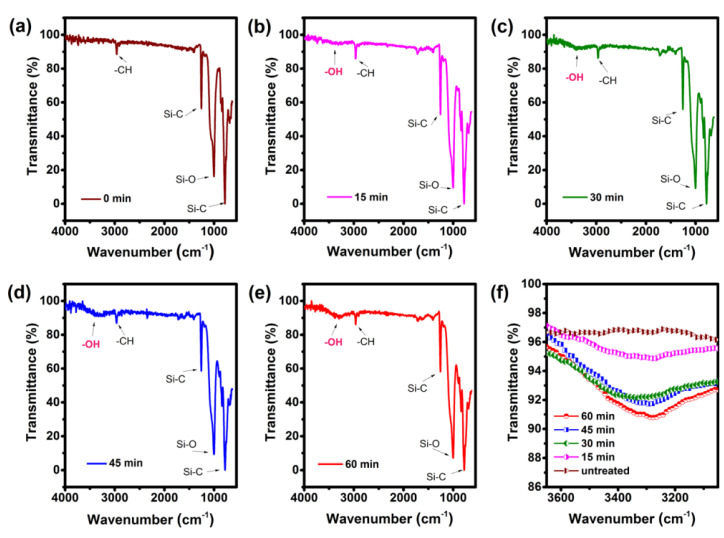
ATR−FTIR spectra of (**a**) untreated PDMS, (**b**) 15 min UVO exposure, (**c**) 30 min UVO exposure, (**d**) 45 min UVO exposure, (**e**) 60 min UVO exposure and (**f**) OH peaks at different UVO exposure times. Reprinted with permission from [94]. Copyright 2018 American Chemical Society.

**Figure 9 polymers-15-01545-f009:**
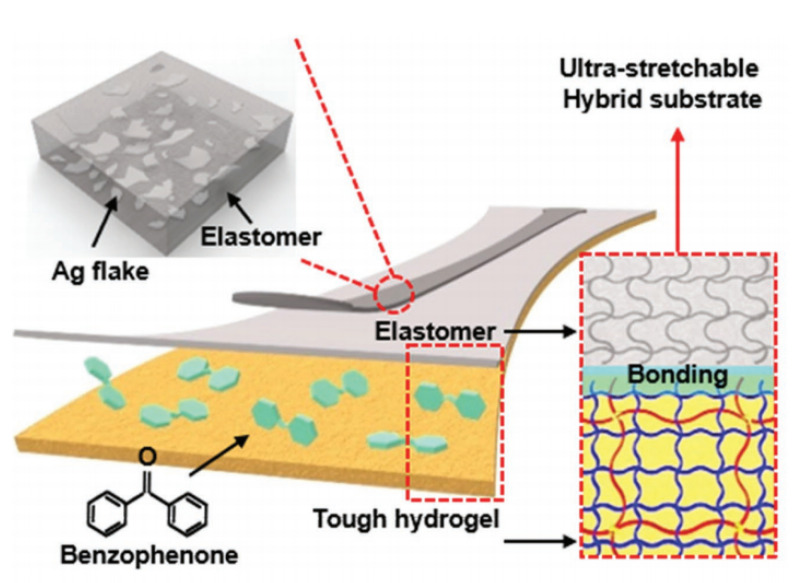
Schematic illustration of the conductor composition. The thin elastomer film is strongly bonded to the tough hydrogel layer by benzophenone. Reprinted with permission from [103]. Copyright 2018 John Wiley and Sons.

**Figure 10 polymers-15-01545-f010:**
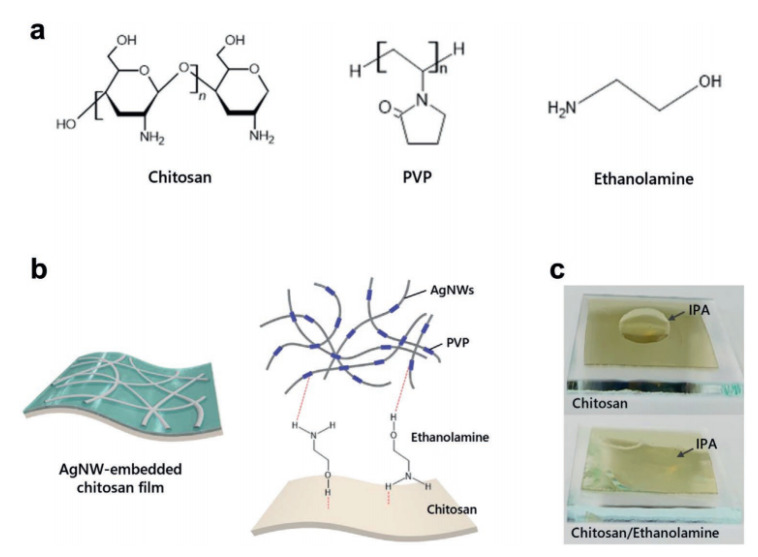
(**a**) Chemical structures of chitosan, PVP and ethanolamine (EA). (**b**) Schematic of the AgNW-embedded chitosan film and interface functionalization mechanism between the AgNWs and the chitosan substrate. (**c**) IPA droplet images of the untreated chitosan and the EA-treated chitosan films. Reprinted with permission from [110]. Copyright 2021 John Wiley and Sons.

**Figure 11 polymers-15-01545-f011:**
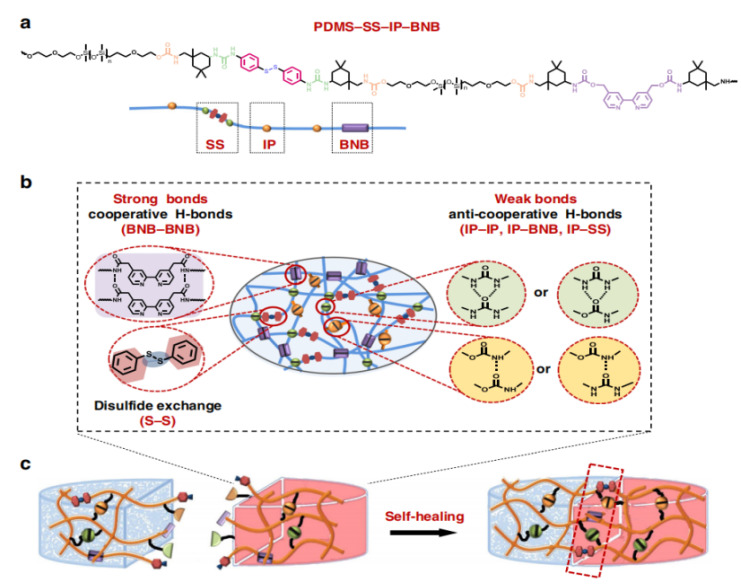
Molecular design of the PDMS-SS-IP-BNB elastomer with high toughness, stretchability and universally autonomous self-healing ability. (**a**) Chemical structure of PDMS-SS-IP-BNB. (**b**) The proposed ideal structure of the supramolecular polymer network based on strong crosslinking H-bonds (BNB-BNB), weak crosslinking H-bonds (IP-IP, IP-BNB, IP-SS) and disulfide metathesis (S-S). (**c**) The synergistic interaction of multiple dynamic bonds contributes to the universally self-healing capability of elastomers. Reprinted with permission from [113]. Copyright 2020 Springer Nature.

**Figure 12 polymers-15-01545-f012:**
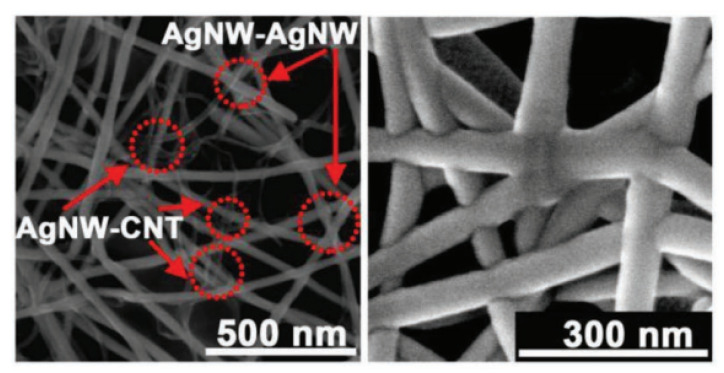
Typical SEM image of AgNW/CNT hybrids after cold welding (**left**) and SEM image of welded AgNW film (**right**). Reprinted with permission from [115]. Copyright 2019 John Wiley and Sons.

**Table 1 polymers-15-01545-t001:** Comparison of different preparation methods.

Method	Maximum Tensile Rate under Conductive Condition or Maximum Conductivity	Ref.	Advantages	Disadvantages
Prestretching/topological matrix	750%	[52]	Simple preparation process, low cost	Generally low tensile rate, estimate deformation direction
	100%	[55]		
Conductive fiber	32.09 S/m	[58]	High conductivity, good tensile properties, can be woven	Process is relatively complex, best to assist with pre-stretching
	500%	[59]		
	200%, 1.4 × 10${}^{6}$ S/m	[60]		
	220%	[61]		
	500%, 6.88 × 10${}^{4}$ S/m	[62]		
	100%	[63]		
Aerogel composites	3 × 10${}^{6}$ S/m	[64]	Light weight, low density	Low tensile property, complex preparation process
	2.1 × 10${}^{5}$ S/m	[65]		
	130%, 6570 S/m	[66]		
	5.1 × 10${}^{4}$ S/m	[67]		
Mixed seepage dopant	200%, 3.668 × 10${}^{5}$ S/m	[70]	High elongation, components are combined, arbitrary structure, good versatility	Complex material ratio, more procedures
	240%	[71]		
	800%	[72]		
	840%, 7.6 × 10${}^{6}$ S/m	[73]		

## Data Availability

Not applicable.

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
