# Peer review of "Silver-Nanowire-Based Elastic Conductors: Preparation Processes and Substrate Adhesion"

_polymers, 2023, doi:10.3390/polym15061545_

Round 1

Reviewer 1 Report

Journal: Polymers (ISSN 2073-4360)

Manuscript ID: polymers-2257830

Type: Article

Title: Silver nanowire based elastic conductors: preparation processes and substrate adhesion.

Authors: Kai Yu , Tian He *.

a)           Write the perspective of the present work carefully.

b)          Do you consider the topic original or relevant in the field? Does it address a specific gap in the field?

c)           Why the authors didn’t measure the optical properties and electrical of the samples?

d)          Refer to the recent refs for the calculations

DOI: https://doi.org/10.1088/1742-6596/1795/1/012052

DOI: https://doi.org/10.1088/1742-6596/1795/1/012059

Best Regards

Author Response

  1. Write the perspective of the present work carefully.

Response: The viewpoints on relevant prior research work have been rewritten prudentially across the whole review.

  1. Do you consider the topic original or relevant in the field? Does it address a specific gap in the field?

Response: As a literature review, we believe the manuscript has its own original and relevant insights into the field of flexible/stretchable conductors. We need more clarification as to what “a specific gap in the field” exactly means.

  1. Why the authors didn’t measure the optical properties and electrical of the samples?

Refer to the recent refs for the calculations

Response: We have noticed that the Type of the manuscript was identified as Article in Reviewer 1’s Comments and Suggestions for Authors, while we submitted a literature review. The request for measurement of optical and electrical properties of the samples seems to originate from a fundamental misclassification of the manuscript, and is completely not applicable to a literature review. Besides, optical properties are not of concern in the discussion of this literature review exclusively focused on conductivity of elastic conductors. The same reason negates the relevancy of the suggested references for optical property calculations.

Reviewer 2 Report

The manuscript " Silver nanowire-based elastic conductors: preparation processes and substrate adhesion" presents a literature review of the manufacturing process of silver nanowire-based elastic conductors. The suggestions for improvement are outlined below.

The manuscript is original and has its merits, but it needs to be improved.

Correct the title. Add a hyphen between nanowire-based.

The abstract does not clearly state the authors' methodology, results, and conclusions.

How many papers were found on the topic?

What keywords were used to search the literature?

A methodology for analyzing the literature must be used.

Page 3 - Figure 1: Improving image quality. It is not possible to read the image scale.

Page 9 - Figure 6: Reduce the size of the figure.

Page 10 – Figure 7: Improve the image quality. It is not possible to read the image scale.

Page 16 - Summary and outlook – To improve the papers, the conclusions need to bring a comparison between the production methods explaining which one is more efficient and bringing some comparison about the adhesion methods. 

Author Response

  1. Correct the title. Add a hyphen between nanowire-based.

Response: Hyphen was added between nanowire and based.

  1. The abstract does not clearly state the authors' methodology, results, and conclusions.

How many papers were found on the topic?

What keywords were used to search the literature?

A methodology for analyzing the literature must be used.

Response: Methodology, results, and conclusions were added to the abstract, last paragragh of introduction, and summary.

  1. Page 3 - Figure 1: Improving image quality. It is not possible to read the image scale.

Page 9 - Figure 6: Reduce the size of the figure.

Page 10 – Figure 7: Improve the image quality. It is not possible to read the image scale.

Response: Page 3 - Figure 1: The size (50μm) has been added in the caption.

Page 9 - Figure 6: The image size has been reduced.

Page 10 – Figure 7: After consulting the original text, the size of the image scale bar is not indicated in the full text as it is non-critical, so the scale bar has been deleted and the image size has been reduced.

Page 16 - Summary and outlook – To improve the papers, the conclusions need to bring a comparison between the production methods explaining which one is more efficient and bringing some comparison about the adhesion methods. 

Response: The comparison between the production methods and the adhesion methods have been added to the summary and outlook.

Reviewer 3 Report

The review by T. He et al. describes the various aspects of preparation and adhesion strategies of Silver nanowire-based elastic conductors via literature examples. A detailed description of four kinds of elastic preparation principles (pre-stretching/geometrically topological matrix, conductive fiber, aerogel composite, mixed percolation dopant) and five kinds of adhesion strategies (coating method, embedding method, changing surface energy, chemical bond and force, adjusting tension and diffusion) are displayed with good examples.

The review is informative and contributes to research in the community of elastic soft polymer electronics material. According to me, the review can be accepted only after answering some of the genuine concerns.

  1. Although the flow of the introduction is good, the authors haven’t done justice to some of the researchers who have contributed to the field (See Journal of Colloid and Interface Science (2018), 527, 315-327, ACS Applied Materials & Interfaces (2021), 13(11), 13705-13713, Applied Physics Letters (2016), 108(12), 121103/1-121103/5, IEEE Sensors Journal (2020), 20(23), 14118-14125, Colloids and Surfaces, A: Physicochemical and Engineering Aspects (2021), 629, 127477). It is suggested to incorporate these references in the review in the corresponding discussion parts.
  2. The figure 4 has a missing Copyright declaration. Please correct it.
  3. A summary table showing the comparison of mechanical and conductivity values on different preparation procedures as well as the advantages and disadvantages of the different methodologies based on literature reports are suggested.
  4. The authors have mentioned the development of these materials in 3D printing and soft robot research. A separate section like ''Applications in 3D printing and soft robotics'' before the summary showing literature examples of the same aspects can be added. (eg. Nanoscale (2018), 10(15), 6806-6811, Advanced Materials (Weinheim, Germany) (2016), 28(28), 5986-5996, Additive Manufacturing (2021), 48(Part_B), 102473, Nano Research (2020), 13(10), 2879-2884, Advanced Materials (Weinheim, Germany) (2016), 28(4), 722-728).

Author Response

  1. Although the flow of the introduction is good, the authors haven’t done justice to some of the researchers who have contributed to the field (See Journal of Colloid and Interface Science (2018), 527, 315-327, ACS Applied Materials & Interfaces (2021), 13(11), 13705-13713, Applied Physics Letters (2016), 108(12), 121103/1-121103/5, IEEE Sensors Journal (2020), 20(23), 14118-14125, Colloids and Surfaces, A: Physicochemical and Engineering Aspects (2021), 629, 127477). It is suggested to incorporate these references in the review in the corresponding discussion parts.

Response: All the above papers have been added to the corresponding parts of the review.

  1. The figure 4 has a missing Copyright declaration. Please correct it.

Response: Copyright declaration has been added.

  1. A summary table showing the comparison of mechanical and conductivity values on different preparation procedures as well as the advantages and disadvantages of the different methodologies based on literature reports are suggested.

Response: The summary table has been added.

  1. The authors have mentioned the development of these materials in 3D printing and soft robot research. A separate section like ''Applications in 3D printing and soft robotics'' before the summary showing literature examples of the same aspects can be added. (eg. Nanoscale (2018), 10(15), 6806-6811, Advanced Materials (Weinheim, Germany) (2016), 28(28), 5986-5996, Additive Manufacturing (2021), 48(Part_B), 102473, Nano Research (2020), 13(10), 2879-2884, Advanced Materials (Weinheim, Germany) (2016), 28(4), 722-728).

Response: The section ''Applications in 3D printing and soft robotics'' has been added.

Round 2

Reviewer 3 Report

The review looks good after the revision, and I would suggest accepting it in the present form.